# Efficient and Robust Automated Machine Learning

**Matthias Feurer**  **Aaron Klein**  **Katharina Eggensperger**
**Jost Tobias Springenberg**  **Manuel Blum**  **Frank Hutter**
Department of Computer Science
University of Freiburg, Germany
{feurerm,kleinaa,eggenspk,springj,mblum,fh}@cs.uni-freiburg.de

## Abstract

The success of machine learning in a broad range of applications has led to an ever-growing demand for machine learning systems that can be used off the shelf by non-experts. To be effective in practice, such systems need to automatically choose a good algorithm and feature preprocessing steps for a new dataset at hand, and also set their respective hyperparameters. Recent work has started to tackle this *automated machine learning (AutoML)* problem with the help of efficient Bayesian optimization methods. Building on this, we introduce a robust new AutoML system based on scikit-learn (using 15 classifiers, 14 feature preprocessing methods, and 4 data preprocessing methods, giving rise to a structured hypothesis space with 110 hyperparameters). This system, which we dub AUTO-SKLEARN, improves on existing AutoML methods by automatically taking into account past performance on similar datasets, and by constructing ensembles from the models evaluated during the optimization. Our system won the first phase of the ongoing ChaLearn AutoML challenge, and our comprehensive analysis on over 100 diverse datasets shows that it substantially outperforms the previous state of the art in AutoML. We also demonstrate the performance gains due to each of our contributions and derive insights into the effectiveness of the individual components of AUTO-SKLEARN.

## 1 Introduction

Machine learning has recently made great strides in many application areas, fueling a growing demand for machine learning systems that can be used effectively by novices in machine learning. Correspondingly, a growing number of commercial enterprises aim to satisfy this demand (*e.g.*, BigML.com, Wise.io, SkyTree.com, RapidMiner.com, Dato.com, Prediction.io, DataRobot.com, Microsoft's Azure Machine Learning, Google's Prediction API, and Amazon Machine Learning). At its core, every effective machine learning service needs to solve the fundamental problems of deciding which machine learning algorithm to use on a given dataset, whether and how to preprocess its features, and how to set all hyperparameters. This is the problem we address in this work.

More specifically, we investigate automated machine learning (AutoML), the problem of automatically (without human input) producing test set predictions for a new dataset within a fixed computational budget. Formally, this AutoML problem can be stated as follows:

**Definition 1** (AutoML problem). *For $i = 1, \ldots, n+m$, let $\boldsymbol{x}_i \in \mathbb{R}^d$ denote a feature vector and $y_i \in Y$ the corresponding target value. Given a training dataset $D_{train} = \{(\boldsymbol{x}_1, y_1), \ldots, (\boldsymbol{x}_n, y_n)\}$ and the feature vectors $\boldsymbol{x}_{n+1}, \ldots, \boldsymbol{x}_{n+m}$ of a test dataset $D_{test} = \{(\boldsymbol{x}_{n+1}, y_{n+1}), \ldots, (\boldsymbol{x}_{n+m}, y_{n+m})\}$ drawn from the same underlying data distribution, as well as a resource budget $b$ and a loss metric $\mathcal{L}(\cdot, \cdot)$, the AutoML problem is to (automatically) produce test set predictions $\hat{y}_{n+1}, \ldots, \hat{y}_{n+m}$. The loss of a solution $\hat{y}_{n+1}, \ldots, \hat{y}_{n+m}$ to the AutoML problem is given by $\frac{1}{m} \sum_{j=1}^{m} \mathcal{L}(\hat{y}_{n+j}, y_{n+j})$.*

In practice, the budget $b$ would comprise computational resources, such as CPU and/or wallclock time and memory usage. This problem definition reflects the setting of the ongoing ChaLearn AutoML challenge [1]. The AutoML system we describe here won the first phase of that challenge.

Here, we follow and extend the AutoML approach first introduced by AUTO-WEKA [2] (see http://automl.org). At its core, this approach combines a highly parametric machine learning framework $F$ with a Bayesian optimization [3] method for instantiating $F$ well for a given dataset.

The contribution of this paper is to extend this AutoML approach in various ways that considerably improve its *efficiency* and *robustness*, based on principles that apply to a wide range of machine learning frameworks (such as those used by the machine learning service providers mentioned above). First, following successful previous work for low dimensional optimization problems [4, 5, 6], we reason across datasets to identify instantiations of machine learning frameworks that perform well on a new dataset and warmstart Bayesian optimization with them (Section 3.1). Second, we automatically construct ensembles of the models considered by Bayesian optimization (Section 3.2). Third, we carefully design a highly parameterized machine learning framework from high-performing classifiers and preprocessors implemented in the popular machine learning framework scikit-learn [7] (Section 4). Finally, we perform an extensive empirical analysis using a diverse collection of datasets to demonstrate that the resulting AUTO-SKLEARN system outperforms previous state-of-the-art AutoML methods (Section 5), to show that each of our contributions leads to substantial performance improvements (Section 6), and to gain insights into the performance of the individual classifiers and preprocessors used in AUTO-SKLEARN (Section 7).

## 2  AutoML as a CASH problem

We first review the formalization of AutoML as a *Combined Algorithm Selection and Hyperparameter optimization (CASH)* problem used by AUTO-WEKA's AutoML approach. Two important problems in AutoML are that (1) no single machine learning method performs best on all datasets and (2) some machine learning methods (*e.g.*, non-linear SVMs) crucially rely on hyperparameter optimization. The latter problem has been successfully attacked using Bayesian optimization [3], which nowadays forms a core component of an AutoML system. The former problem is intertwined with the latter since the rankings of algorithms depend on whether their hyperparameters are tuned properly. Fortunately, the two problems can efficiently be tackled as a single, structured, joint optimization problem:

**Definition 2** (CASH). *Let $\mathcal{A} = \{A^{(1)}, \ldots, A^{(R)}\}$ be a set of algorithms, and let the hyperparameters of each algorithm $A^{(j)}$ have domain $\Lambda^{(j)}$. Further, let $D_{train} = \{(x_1, y_1), \ldots, (x_n, y_n)\}$ be a training set which is split into $K$ cross-validation folds $\{D_{valid}^{(1)}, \ldots, D_{valid}^{(K)}\}$ and $\{D_{train}^{(1)}, \ldots, D_{train}^{(K)}\}$ such that $D_{train}^{(i)} = D_{train} \backslash D_{valid}^{(i)}$ for $i = 1, \ldots, K$. Finally, let $\mathcal{L}(A_{\boldsymbol{\lambda}}^{(j)}, D_{train}^{(i)}, D_{valid}^{(i)})$ denote the loss that algorithm $A^{(j)}$ achieves on $D_{valid}^{(i)}$ when trained on $D_{train}^{(i)}$ with hyperparameters $\boldsymbol{\lambda}$. Then, the* Combined Algorithm Selection and Hyperparameter optimization (CASH) *problem is to find the joint algorithm and hyperparameter setting that minimizes this loss:*

$$A^{\star}, \boldsymbol{\lambda}_{\star} \in \underset{A^{(j)} \in \mathcal{A}, \boldsymbol{\lambda} \in \Lambda^{(j)}}{\operatorname{argmin}} \frac{1}{K} \sum_{i=1}^{K} \mathcal{L}(A_{\boldsymbol{\lambda}}^{(j)}, D_{train}^{(i)}, D_{valid}^{(i)}). \tag{1}$$

This CASH problem was first tackled by Thornton et al. [2] in the AUTO-WEKA system using the machine learning framework WEKA [8] and tree-based Bayesian optimization methods [9, 10]. In a nutshell, Bayesian optimization [3] fits a probabilistic model to capture the relationship between hyperparameter settings and their measured performance; it then uses this model to select the most promising hyperparameter setting (trading off exploration of new parts of the space *vs.* exploitation in known good regions), evaluates that hyperparameter setting, updates the model with the result, and iterates. While Bayesian optimization based on Gaussian process models (*e.g.*, Snoek et al. [11]) performs best in low-dimensional problems with numerical hyperparameters, tree-based models have been shown to be more successful in high-dimensional, structured, and partly discrete problems [12] – such as the CASH problem – and are also used in the AutoML system HYPEROPT-SKLEARN [13]. Among the tree-based Bayesian optimization methods, Thornton et al. [2] found the random-forest-based SMAC [9] to outperform the tree Parzen estimator TPE [10], and we therefore use SMAC to solve the CASH problem in this paper. Next to its use of random forests [14], SMAC's main distinguishing feature is that it allows fast cross-validation by evaluating one fold at a time and discarding poorly-performing hyperparameter settings early.

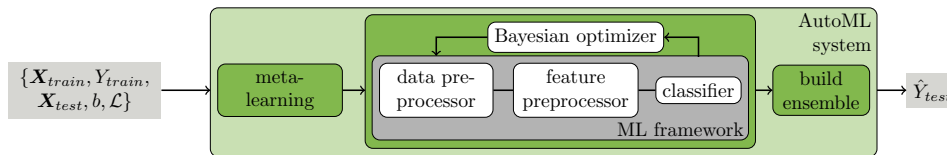

Figure 1: Our improved AutoML approach. We add two components to Bayesian hyperparameter optimization of an ML framework: meta-learning for initializing the Bayesian optimizer and automated ensemble construction from configurations evaluated during optimization.

## 3 New methods for increasing efficiency and robustness of AutoML

We now discuss our two improvements of the AutoML approach. First, we include a meta-learning step to warmstart the Bayesian optimization procedure, which results in a considerable boost in efficiency. Second, we include an automated ensemble construction step, allowing us to use all classifiers that were found by Bayesian optimization.

Figure 1 summarizes the overall AutoML workflow, including both of our improvements. We note that we expect their effectiveness to be greater for flexible ML frameworks that offer many degrees of freedom (*e.g.*, many algorithms, hyperparameters, and preprocessing methods).

### 3.1 Meta-learning for finding good instantiations of machine learning frameworks

Domain experts derive knowledge from previous tasks: They *learn about the performance of machine learning algorithms*. The area of meta-learning [15] mimics this strategy by reasoning about the performance of learning algorithms across datasets. In this work, we apply meta-learning to select instantiations of our given machine learning framework that are likely to perform well on a new dataset. More specifically, for a large number of datasets, we collect both performance data and a set of *meta-features*, i.e., characteristics of the dataset that can be computed efficiently and that help to determine which algorithm to use on a new dataset.

This meta-learning approach is complementary to Bayesian optimization for optimizing an ML framework. Meta-learning can quickly suggest some instantiations of the ML framework that are likely to perform quite well, but it is unable to provide fine-grained information on performance. In contrast, Bayesian optimization is slow to start for hyperparameter spaces as large as those of entire ML frameworks, but can fine-tune performance over time. We exploit this complementarity by selecting $k$ configurations based on meta-learning and use their result to seed Bayesian optimization. This approach of warmstarting optimization by meta-learning has already been successfully applied before [4, 5, 6], but never to an optimization problem as complex as that of searching the space of instantiations of a full-fledged ML framework. Likewise, learning across datasets has also been applied in collaborative Bayesian optimization methods [16, 17]; while these approaches are promising, they are so far limited to very few meta-features and cannot yet cope with the high-dimensional partially discrete configuration spaces faced in AutoML.

More precisely, our meta-learning approach works as follows. In an offline phase, for each machine learning dataset in a dataset repository (in our case 140 datasets from the OpenML [18] repository), we evaluated a set of meta-features (described below) and used Bayesian optimization to determine and store an instantiation of the given ML framework with strong empirical performance for that dataset. (In detail, we ran SMAC [9] for 24 hours with 10-fold cross-validation on two thirds of the data and stored the resulting ML framework instantiation which exhibited best performance on the remaining third). Then, given a new dataset $\mathcal{D}$, we compute its meta-features, rank all datasets by their $L_1$ distance to $\mathcal{D}$ in meta-feature space and select the stored ML framework instantiations for the $k = 25$ nearest datasets for evaluation before starting Bayesian optimization with their results.

To characterize datasets, we implemented a total of 38 meta-features from the literature, including simple, information-theoretic and statistical meta-features [19, 20], such as statistics about the number of data points, features, and classes, as well as data skewness, and the entropy of the targets. All meta-features are listed in Table 1 of the supplementary material. Notably, we had to exclude the prominent and effective category of landmarking meta-features [21] (which measure the performance of simple base learners), because they were computationally too expensive to be helpful in the online evaluation phase. We note that this meta-learning approach draws its power from the availability of

a repository of datasets; due to recent initiatives, such as OpenML [18], we expect the number of available datasets to grow ever larger over time, increasing the importance of meta-learning.

## 3.2 Automated ensemble construction of models evaluated during optimization

While Bayesian hyperparameter optimization is data-efficient in finding the best-performing hyperparameter setting, we note that it is a very wasteful procedure when the goal is simply to make good predictions: all the models it trains during the course of the search are lost, usually including some that perform almost as well as the best. Rather than discarding these models, we propose to store them and to use an efficient post-processing method (which can be run in a second process on-the-fly) to construct an ensemble out of them. This automatic ensemble construction avoids to commit itself to a single hyperparameter setting and is thus more robust (and less prone to overfitting) than using the point estimate that standard hyperparameter optimization yields. To our best knowledge, we are the first to make this simple observation, which can be applied to improve any Bayesian hyperparameter optimization method.

It is well known that ensembles often outperform individual models [22, 23], and that effective ensembles can be created from a library of models [24, 25]. Ensembles perform particularly well if the models they are based on (1) are individually strong and (2) make uncorrelated errors [14]. Since this is much more likely when the individual models are different in nature, ensemble building is particularly well suited for combining strong instantiations of a flexible ML framework.

However, simply building a uniformly weighted ensemble of the models found by Bayesian optimization does *not* work well. Rather, we found it crucial to adjust these weights using the predictions of all individual models on a hold-out set. We experimented with different approaches to optimize these weights: *stacking* [26], gradient-free numerical optimization, and the method *ensemble selection* [24]. While we found both numerical optimization and stacking to overfit to the validation set and to be computationally costly, ensemble selection was fast and robust. In a nutshell, ensemble selection (introduced by Caruana et al. [24]) is a greedy procedure that starts from an empty ensemble and then iteratively adds the model that maximizes ensemble validation performance (with uniform weight, but allowing for repetitions). Procedure 1 in the supplementary material describes it in detail. We used this technique in all our experiments – building an ensemble of size 50.

## 4 A practical automated machine learning system

To design a robust AutoML system, as our underlying ML framework we chose scikit-learn [7], one of the best known and most widely used machine learning libraries. It offers a wide range of well established and efficiently-implemented ML algorithms and is easy to use for both experts and beginners. Since our AutoML system closely resembles AUTO-WEKA, but – like HYPEROPT-SKLEARN – is based on scikit-learn, we dub it AUTO-SKLEARN.

Figure 2 depicts AUTO-SKLEARN's overall components. It comprises 15 classification algorithms, 14 preprocessing methods, and 4 data preprocessing methods. We parameterized each of them, which resulted in a

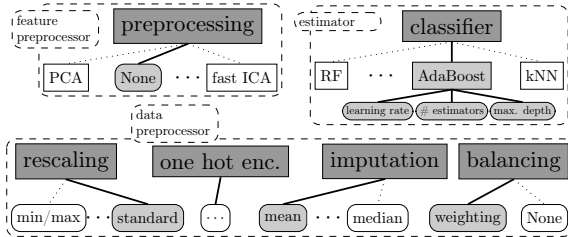

Figure 2: Structured configuration space. Squared boxes denote parent hyperparameters whereas boxes with rounded edges are leaf hyperparameters. Grey colored boxes mark active hyperparameters which form an example configuration and machine learning pipeline. Each pipeline comprises one *feature preprocessor*, *classifier* and up to three *data preprocessor* methods plus respective hyperparameters.

space of 110 hyperparameters. Most of these are conditional hyperparameters that are only active if their respective component is selected. We note that SMAC [9] can handle this conditionality natively.

All 15 classification algorithms in AUTO-SKLEARN are listed in Table 1a (and described in detail in Section A.1 of the supplementary material). They fall into different categories, such as general linear models (2 algorithms), support vector machines (2), discriminant analysis (2), nearest neighbors (1), naïve Bayes (3), decision trees (1) and ensembles (4). In contrast to AUTO-WEKA [2], we

| name | #λ | cat (cond) | cont (cond) |
|---|---|---|---|
| AdaBoost (AB) | 4 | 1 (-) | 3 (-) |
| Bernoulli naïve Bayes | 2 | 1 (-) | 1 (-) |
| decision tree (DT) | 4 | 1 (-) | 3 (-) |
| extreml. rand. trees | 5 | 2 (-) | 3 (-) |
| Gaussian naïve Bayes | - | - | - |
| gradient boosting (GB) | 6 | - | 6 (-) |
| kNN | 3 | 2 (-) | 1 (-) |
| LDA | 4 | 1 (-) | 3 (1) |
| linear SVM | 4 | 2 (-) | 2 (-) |
| kernel SVM | 7 | 2 (-) | 5 (2) |
| multinomial naïve Bayes | 2 | 1 (-) | 1 (-) |
| passive aggressive | 3 | 1 (-) | 2 (-) |
| QDA | 2 | - | 2 (-) |
| random forest (RF) | 5 | 2 (-) | 3 (-) |
| Linear Class. (SGD) | 10 | 4 (-) | 6 (3) |

(a) classification algorithms

| name | #λ | cat (cond) | cont (cond) |
|---|---|---|---|
| extreml. rand. trees prepr. | 5 | 2 (-) | 3 (-) |
| fast ICA | 4 | 3 (-) | 1 (1) |
| feature agglomeration | 4 | 3 () | 1 (-) |
| kernel PCA | 5 | 1 (-) | 4 (3) |
| rand. kitchen sinks | 2 | - | 2 (-) |
| linear SVM prepr. | 3 | 1 (-) | 2 (-) |
| no preprocessing | - | - | - |
| nystroem sampler | 5 | 1 (-) | 4 (3) |
| PCA | 2 | 1 (-) | 1 (-) |
| polynomial | 3 | 2 (-) | 1 (-) |
| random trees embed. | 4 | - | 4 (-) |
| select percentile | 2 | 1 (-) | 1 (-) |
| select rates | 3 | 2 (-) | 1 (-) |
| one-hot encoding | 2 | 1 (-) | 1 (1) |
| imputation | 1 | 1 (-) | - |
| balancing | 1 | 1 (-) | - |
| rescaling | 1 | 1 (-) | - |

(b) preprocessing methods

Table 1: Number of hyperparameters for each possible classifier (left) and feature preprocessing method (right) for a **binary classification** dataset in **dense** representation. Tables for sparse binary classification and sparse/dense multiclass classification datasets can be found in the Section E of the supplementary material, Tables 2a, 3a, 4a, 2b, 3b and 4b. We distinguish between categorical (cat) hyperparameters with discrete values and continuous (cont) numerical hyperparameters. Numbers in brackets are conditional hyperparameters, which are only relevant when another parameter has a certain value.

focused our configuration space on base classifiers and excluded meta-models and ensembles that are themselves parameterized by one or more base classifiers. While such ensembles increased AUTO-WEKA's number of hyperparameters by almost a factor of five (to 786), AUTO-SKLEARN "only" features 110 hyperparameters. We instead construct complex ensembles using our post-hoc method from Section 3.2. Compared to AUTO-WEKA, this is much more data-efficient: in AUTO-WEKA, evaluating the performance of an ensemble with 5 components requires the construction and evaluation of 5 models; in contrast, in AUTO-SKLEARN, ensembles come largely for free, and it is possible to mix and match models evaluated at arbitrary times during the optimization.

The preprocessing methods for datasets in dense representation in AUTO-SKLEARN are listed in Table 1b (and described in detail in Section A.2 of the supplementary material). They comprise data preprocessors (which change the feature values and are always used when they apply) and feature preprocessors (which change the actual set of features, and only one of which [or none] is used). Data preprocessing includes rescaling of the inputs, imputation of missing values, one-hot encoding and balancing of the target classes. The 14 possible feature preprocessing methods can be categorized into feature selection (2), kernel approximation (2), matrix decomposition (3), embeddings (1), feature clustering (1), polynomial feature expansion (1) and methods that use a classifier for feature selection (2). For example, $L_1$-regularized linear SVMs fitted to the data can be used for feature selection by eliminating features corresponding to zero-valued model coefficients.

As with every robust real-world system, we had to handle many more important details in AUTO-SKLEARN; we describe these in Section B of the supplementary material.

## 5   Comparing AUTO-SKLEARN to AUTO-WEKA and HYPEROPT-SKLEARN

As a baseline experiment, we compared the performance of vanilla AUTO-SKLEARN (without our improvements) to AUTO-WEKA and HYPEROPT-SKLEARN, reproducing the experimental setup with 21 datasets of the paper introducing AUTO-WEKA [2]. We describe this setup in detail in Section G in the supplementary material.

Table 2 shows that AUTO-SKLEARN performed statistically significantly better than AUTO-WEKA in 6/21 cases, tied it in 12 cases, and lost against it in 3. For the three datasets where AUTO-WEKA performed best, we found that in more than 50% of its runs the best classifier it chose is not implemented in scikit-learn (trees with a pruning component). So far, HYPEROPT-SKLEARN is more of a proof-of-concept – inviting the user to adapt the configuration space to her own needs – than a full AutoML system. The current version crashes when presented with sparse data and missing values. It also crashes on Cifar-10 due to a memory limit which we set for all optimizers to enable a

|  | Abalone | Amazon | Car | Cifar-10 | Cifar-10 Small | Convex | Dexter | Dorothea | German Credit | Gisette | KDD09 Appetency | KR-vs-KP | Madelon | MNIST Basic | MRBI | Secom | Semeion | Shuttle | Waveform | Wine Quality | Yeast |
|---|---|---|---|---|---|---|---|---|---|---|---|---|---|---|---|---|---|---|---|---|---|
| AS | **73.50** | **16.00** | 0.39 | **51.70** | **54.81** | **17.53** | **5.56** | **5.51** | **27.00** | **1.62** | **1.74** | 0.42 | **12.44** | 2.84 | **46.92** | **7.87** | **5.24** | **0.01** | 14.93 | 33.76 | 40.67 |
| AW | **73.50** | 30.00 | **0.00** | 56.95 | 56.20 | 21.80 | 8.33 | 6.38 | 28.33 | 2.29 | **1.74** | **0.31** | 18.21 | 2.84 | 60.34 | 8.09 | **5.24** | **0.01** | 14.13 | 33.36 | **37.75** |
| HS | 76.21 | 16.22 | 0.39 | - | 57.95 | 19.18 | - | - | 27.67 | 2.29 | - | 0.42 | 14.74 | **2.82** | 55.79 | - | 5.87 | 0.05 | **14.07** | 34.72 | 38.45 |

Table 2: Test set classification error of AUTO-WEKA (AW), vanilla AUTO-SKLEARN (AS) and HYPEROPT-SKLEARN (HS), as in the original evaluation of AUTO-WEKA [2]. We show median percent error across 100 000 bootstrap samples (based on 10 runs), simulating 4 parallel runs. Bold numbers indicate the best result. Underlined results are not statistically significantly different from the best according to a bootstrap test with $p = 0.05$.

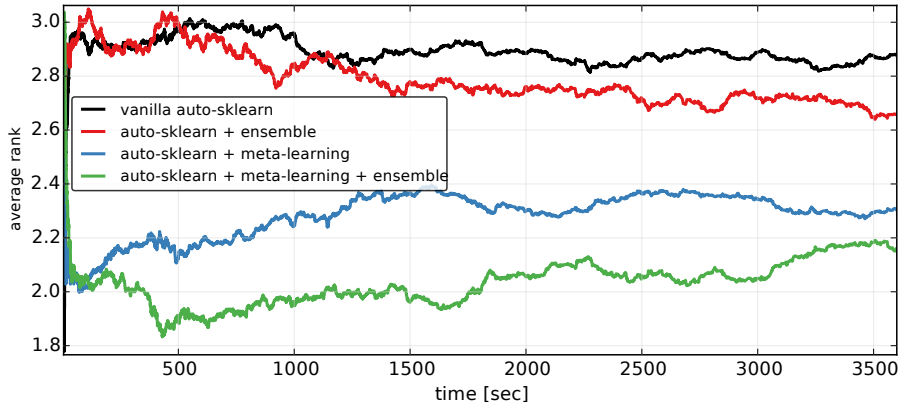

Figure 3: Average rank of all four AUTO-SKLEARN variants (ranked by balanced test error rate (BER)) across 140 datasets. Note that ranks are a relative measure of performance (here, the rank of all methods has to add up to 10), and hence an improvement in BER of one method can worsen the rank of another. The supplementary material shows the same plot on a log-scale to show the time overhead of meta-feature and ensemble computation.

fair comparison. On the 16 datasets on which it ran, it statistically tied the best optimizer in 9 cases and lost against it in 7.

## 6 Evaluation of the proposed AutoML improvements

In order to evaluate the robustness and general applicability of our proposed AutoML system on a broad range of datasets, we gathered 140 binary and multiclass classification datasets from the OpenML repository [18], only selecting datasets with at least 1000 data points to allow robust performance evaluations. These datasets cover a diverse range of applications, such as text classification, digit and letter recognition, gene sequence and RNA classification, advertisement, particle classification for telescope data, and cancer detection in tissue samples. We list all datasets in Table 7 and 8 in the supplementary material and provide their unique OpenML identifiers for reproducibility. Since the class distribution in many of these datasets is quite imbalanced we evaluated all AutoML methods using a measure called *balanced classification error rate* (BER). We define balanced error rate as the average of the proportion of wrong classifications in each class. In comparison to standard classification error (the average overall error), this measure (the average of the *class-wise* error) assigns equal weight to all classes. We note that balanced error or accuracy measures are often used in machine learning competitions (*e.g.*, the AutoML challenge [1] uses balanced accuracy).

We performed 10 runs of AUTO-SKLEARN both with and without meta-learning and with and without ensemble prediction on each of the datasets. To study their performance under rigid time constraints, and also due to computational resource constraints, we limited the CPU time for each run to 1 hour; we also limited the runtime for a single model to a tenth of this (6 minutes). To not evaluate performance on data sets already used for meta-learning, we performed a leave-one-dataset-out validation: when evaluating on dataset $\mathcal{D}$, we only used meta-information from the 139 other datasets.

Figure 3 shows the average ranks over time of the four AUTO-SKLEARN versions we tested. We observe that both of our new methods yielded substantial improvements over vanilla AUTO-SKLEARN. The most striking result is that meta-learning yielded drastic improvements starting with the first

| OpenML dataset ID | AUTO-SKLEARN | AdaBoost | Bernoulli naïve Bayes | decision tree | extreml. rand. trees | Gaussian naïve Bayes | gradient boosting | kNN | LDA | linear SVM | kernel SVM | multinomial naïve Bayes | passive aggressive | QDA | random forest | Linear Class. (SGD) |
|---|---|---|---|---|---|---|---|---|---|---|---|---|---|---|---|---|
| 38 | 2.15 | 2.68 | 50.22 | 2.15 | 18.06 | 11.22 | **1.77** | 50.00 | 8.55 | 16.29 | 17.89 | 46.99 | 50.00 | 8.78 | 2.34 | 15.82 |
| 46 | 3.76 | 4.65 | - | 5.62 | 4.74 | 7.88 | **3.49** | 7.57 | 8.67 | 8.31 | 5.36 | 7.55 | 9.23 | 7.57 | 4.20 | 7.31 |
| 179 | **16.99** | 17.03 | 19.27 | 18.31 | 17.09 | 21.77 | 17.00 | 22.23 | 18.93 | 17.30 | 17.57 | 18.97 | 22.29 | 19.06 | 17.24 | 17.01 |
| 184 | **10.32** | 10.52 | - | 17.46 | 11.10 | 64.74 | 10.42 | 31.10 | 35.44 | 15.76 | 12.52 | 27.13 | 20.01 | 47.18 | 10.98 | 12.76 |
| 554 | 1.55 | 2.42 | - | 12.00 | 2.91 | 10.52 | 3.86 | 2.68 | 3.34 | 2.23 | **1.50** | 10.37 | 100.00 | 2.75 | 3.08 | 2.50 |
| 772 | 46.85 | 49.68 | 47.90 | 47.75 | **45.62** | 48.83 | 48.15 | 48.00 | 46.74 | 48.38 | 48.66 | 47.21 | 48.75 | 47.67 | 47.71 | 47.93 |
| 917 | 10.22 | 9.11 | 25.83 | 11.00 | 10.22 | 33.94 | 10.11 | 11.11 | 34.22 | 18.67 | **6.78** | 25.50 | 20.67 | 30.44 | 10.83 | 18.33 |
| 1049 | 12.93 | **12.53** | 15.50 | 19.31 | 17.18 | 26.23 | 13.38 | 23.80 | 25.12 | 17.28 | 21.44 | 26.40 | 29.25 | 21.38 | 13.75 | 19.92 |
| 1111 | 23.70 | 23.16 | 28.40 | 24.40 | 24.47 | 29.59 | **22.93** | 50.30 | 24.11 | 23.99 | 23.56 | 27.67 | 43.79 | 25.86 | 28.06 | 23.36 |
| 1120 | 13.81 | **13.54** | 18.81 | 17.45 | 13.86 | 21.50 | 13.61 | 17.23 | 15.48 | 14.94 | 14.17 | 18.33 | 16.37 | 15.62 | 13.70 | 14.66 |
| 1128 | 4.21 | 4.89 | 4.71 | 9.30 | 3.89 | 4.77 | 4.58 | 4.59 | 4.58 | 4.83 | 4.59 | 4.46 | 5.65 | 5.59 | **3.83** | 4.33 |
| 293 | 2.86 | 4.07 | 24.30 | 5.03 | 3.59 | 32.44 | 24.48 | 4.86 | 24.40 | 14.16 | 100.00 | 24.20 | 21.34 | 28.68 | **2.57** | 15.54 |
| 389 | 19.65 | 22.98 | - | 33.14 | 19.38 | 29.18 | 19.20 | 30.87 | 19.68 | **17.95** | 22.04 | 20.04 | 20.14 | 39.57 | 20.66 | 17.99 |

Table 3: Median balanced test error rate (BER) of optimizing AUTO-SKLEARN subspaces for each classification method (and all preprocessors), as well as the whole configuration space of AUTO-SKLEARN, on 13 datasets. All optimization runs were allowed to run for 24 hours except for AUTO-SKLEARN which ran for 48 hours. Bold numbers indicate the best result; underlined results are not statistically significantly different from the best according to a bootstrap test using the same setup as for Table 2.

| OpenML dataset ID | AUTO-SKLEARN | densifier | extreml. rand. trees prepr. | fast ICA | feature agglomeration | kernel PCA | rand. kitchen sinks | linear SVM prepr. | no preproc. | nystroem sampler | PCA | polynomial | random trees embed. | select percentile classification | select rates | truncatedSVD |
|---|---|---|---|---|---|---|---|---|---|---|---|---|---|---|---|---|
| 38 | **2.15** | - | 4.03 | 7.27 | 2.24 | 5.84 | 8.57 | 2.28 | 2.28 | 7.70 | 7.23 | 2.90 | 18.50 | 2.20 | 2.28 | - |
| 46 | **3.76** | - | 4.98 | 7.95 | 4.40 | 8.74 | 8.41 | 4.25 | 4.52 | 8.48 | 8.40 | 4.21 | 7.51 | 4.17 | 4.68 | - |
| 179 | 16.99 | - | 17.83 | 17.24 | 16.92 | 100.00 | 17.34 | **16.84** | 16.97 | 17.30 | 17.64 | 16.94 | 17.05 | 17.09 | 16.86 | - |
| 184 | 10.32 | - | 55.78 | 19.96 | 11.31 | 36.52 | 28.05 | **9.92** | 11.43 | 25.53 | 21.15 | 10.54 | 12.68 | 45.03 | 10.47 | - |
| 554 | 1.55 | - | 1.56 | 2.52 | 1.65 | 100.00 | 100.00 | 2.21 | 1.60 | 2.21 | 1.65 | 100.00 | 3.48 | **1.46** | 1.70 | - |
| 772 | **46.85** | - | 47.90 | 48.65 | 48.62 | 47.59 | 47.68 | 47.72 | 48.34 | 48.06 | 47.30 | 48.00 | 47.84 | 47.56 | 48.43 | - |
| 917 | 10.22 | - | **8.33** | 16.06 | 10.33 | 20.94 | 35.44 | 8.67 | 9.44 | 37.83 | 22.33 | 9.11 | 17.67 | 10.00 | 10.44 | - |
| 1049 | 12.93 | - | 20.36 | 19.92 | 13.14 | 19.57 | 20.06 | 13.28 | 15.84 | 18.96 | 17.22 | 12.95 | 18.52 | **11.94** | 14.38 | - |
| 1111 | 23.70 | - | 23.36 | 24.69 | 23.73 | 100.00 | 25.25 | 23.43 | **22.27** | 23.95 | 23.25 | 26.94 | 26.68 | 23.53 | 23.33 | - |
| 1120 | 13.81 | - | 16.29 | 14.22 | 13.73 | 14.57 | 14.82 | 14.02 | 13.85 | 14.66 | 14.23 | **13.22** | 15.03 | 13.65 | 13.67 | - |
| 1128 | 4.21 | - | 4.90 | 4.96 | 4.76 | 4.21 | 5.08 | 4.52 | 4.59 | **4.08** | 4.59 | 50.00 | 9.23 | 4.33 | **4.08** | - |
| 293 | 2.86 | 24.40 | 3.41 | - | - | 100.00 | 19.30 | 3.01 | **2.66** | 20.94 | - | - | 8.05 | 2.86 | 2.74 | 4.05 |
| 389 | 19.65 | 20.63 | 21.40 | - | - | **17.50** | 19.66 | 19.89 | 20.87 | 18.46 | - | - | 44.83 | 20.17 | 19.18 | 21.58 |

Table 4: Like Table 3, but instead optimizing subspaces for each preprocessing method (and all classifiers).

configuration it selected and lasting until the end of the experiment. We note that the improvement was most pronounced in the beginning and that over time, vanilla AUTO-SKLEARN also found good solutions without meta-learning, letting it catch up on some datasets (thus improving its overall rank).

Moreover, both of our methods complement each other: our automated ensemble construction improved both vanilla AUTO-SKLEARN and AUTO-SKLEARN with meta-learning. Interestingly, the ensemble's influence on the performance started earlier for the meta-learning version. We believe that this is because meta-learning produces better machine learning models earlier, which can be directly combined into a strong ensemble; but when run longer, vanilla AUTO-SKLEARN without meta-learning also benefits from automated ensemble construction.

# 7 Detailed analysis of AUTO-SKLEARN components

We now study AUTO-SKLEARN's individual classifiers and preprocessors, compared to jointly optimizing all methods, in order to obtain insights into their peak performance and robustness. Ideally, we would have liked to study all combinations of a single classifier and a single preprocessor in isolation, but with 15 classifiers and 14 preprocessors this was infeasible; rather, when studying the performance of a single classifier, we still optimized over all preprocessors, and vice versa. To obtain a more detailed analysis, we focused on a subset of datasets but extended the configuration budget for optimizing all methods from one hour to one day and to two days for AUTO-SKLEARN. Specifically, we clustered our 140 datasets with g-means [27] based on the dataset meta-features and used one dataset from each of the resulting 13 clusters (see Table 6 in the supplementary material for the list of datasets). We note that, in total, these extensive experiments required 10.7 CPU years.

Table 3 compares the results of the various classification methods against AUTO-SKLEARN. Overall, as expected, random forests, extremely randomized trees, AdaBoost, and gradient boosting, showed

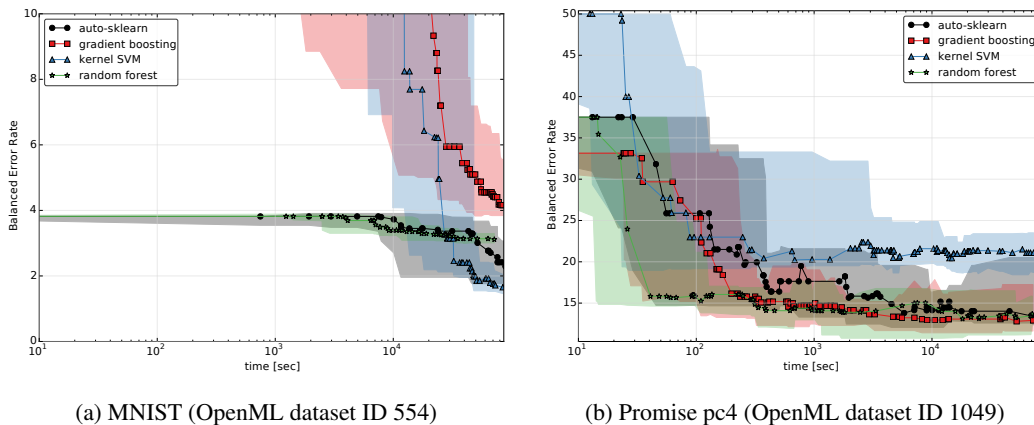

(a) MNIST (OpenML dataset ID 554)  (b) Promise pc4 (OpenML dataset ID 1049)

Figure 4: Performance of a subset of classifiers compared to AUTO-SKLEARN over time. We show median test error rate and the fifth and 95th percentile over time for optimizing three classifiers separately with optimizing the joint space. A plot with all classifiers can be found in Figure 4 in the supplementary material. While AUTO-SKLEARN is inferior in the beginning, in the end its performance is close to the best method.

the most robust performance, and SVMs showed strong peak performance for some datasets. Besides a variety of strong classifiers, there are also several models which could not compete: The decision tree, passive aggressive, kNN, Gaussian NB, LDA and QDA were statistically significantly inferior to the best classifier on most datasets. Finally, the table indicates that no single method was the best choice for all datasets. As shown in the table and also visualized for two example datasets in Figure 4, optimizing the joint configuration space of AUTO-SKLEARN led to the most robust performance. A plot of ranks over time (Figure 2 and 3 in the supplementary material) quantifies this across all 13 datasets, showing that AUTO-SKLEARN starts with reasonable but not optimal performance and effectively searches its more general configuration space to converge to the best overall performance over time.

Table 4 compares the results of the various preprocessors against AUTO-SKLEARN. As for the comparison of classifiers above, AUTO-SKLEARN showed the most robust performance: It performed best on three of the datasets and was not statistically significantly worse than the best preprocessor on another 8 of 13.

# 8 Discussion and Conclusion

We demonstrated that our new AutoML system AUTO-SKLEARN performs favorably against the previous state of the art in AutoML, and that our meta-learning and ensemble improvements for AutoML yield further efficiency and robustness. This finding is backed by the fact that AUTO-SKLEARN won the auto-track in the first phase of ChaLearn's ongoing AutoML challenge. In this paper, we did not evaluate the use of AUTO-SKLEARN for interactive machine learning with an expert in the loop and weeks of CPU power, but we note that that mode has also led to a third place in the human track of the same challenge. As such, we believe that AUTO-SKLEARN is a promising system for use by both machine learning novices and experts. The source code of AUTO-SKLEARN is available under an open source license at `https://github.com/automl/auto-sklearn`.

Our system also has some shortcomings, which we would like to remove in future work. As one example, we have not yet tackled regression or semi-supervised problems. Most importantly, though, the focus on scikit-learn implied a focus on small to medium-sized datasets, and an obvious direction for future work will be to apply our methods to modern deep learning systems that yield state-of-the-art performance on large datasets; we expect that in that domain especially automated ensemble construction will lead to tangible performance improvements over Bayesian optimization.

## Acknowledgments

This work was supported by the German Research Foundation (DFG), under Priority Programme Autonomous Learning (SPP 1527, grant HU 1900/3-1), under Emmy Noether grant HU 1900/2-1, and under the BrainLinks-BrainTools Cluster of Excellence (grant number EXC 1086).

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
