[Supplementary Material · nips2015_supplementary.pdf]

# Supplementary Material for Efficient and Robust Automated Machine Learning

**Matthias Feurer**          **Aaron Klein**          **Katharina Eggensperger**
**Jost Tobias Springenberg**          **Manuel Blum**          **Frank Hutter**
Department of Computer Science
University of Freiburg, Germany
{feurerm,kleinaa,eggenspk,springj,mblum,fh}@cs.uni-freiburg.de

## A  Description of the classification algorithms and preprocessing methods

In this section we give a more detailed explanation of the classification and preprocessing methods that we used in AUTO-SKLEARN.

### A.1  Classification algorithms

Our AUTO-SKLEARN framework contains 15 base classifiers from scikit learn (out of which exactly one is chosen at each point during the optimization process). The 15 algorithms can generally be separated into 7 categories: generalized linear models (2 algorithms), support vector machines (2), discriminant analysis (2), nearest neighbors (1), naïve Bayes (3), decision trees (1) and ensemble methods (4). A complete list of the algorithms is given in Table 1a in the main paper. While an in-depth description of each algorithm is out of the scope of this paper we want to give a brief description of each category and highlight complementary strengths of algorithms within one category.

**Generalized linear models.**   The first class of algorithms we consider are generalized linear models (GLM) for classification. These are linear classification algorithms. Since we are interested in scaling our AutoML system to medium to large datasets we only use online learning algorithms from this category: Linear Classification via online stochastic gradient descent (SGD) either with a negative log likelihood, a hinge or a Huber los, and maximum margin classification via online passive aggressive algorithms [1] – which iteratively solve constrained optimization problems to update the model weights to both guarantee small steps and retain a large margin.

**Support vector machines.**   Closely related to the algorithms from the GLM class described above support vector machines construct a maximal margin separating hyperplane by minimizing the hinge loss on the training data. As is well known, they can also be used for non-linear classification by employing the "kernel trick". The SVM implementations used in scikit-learn are based on online optimization using LibSVM [2] or liblinear [3] as backends.

**Discriminant analysis.**   We also consider two instantiations from the family of discriminant analysis methods: (1) Quadratic discriminant analysis (QDA) assumes that the feature values for each class are normally distributed. Classification is done by applying the likelihood ratio test. (2) Linear discriminant analysis (LDA) makes the additional assumption that the covariance of each of the classes is identical, which leads to a linear decision boundary.

**Nearest neighbors classification.**   k-nearest neighbors is a non-parametric classification algorithm that classifies samples based on the class membership of their nearest neighbors in feature space. Nearest neighbor classifiers often exhibit strong performance in problems where a proper metric in feature space is known, but can be computationally expensive to compute for large datasets (when using a basic implementation as that contained in scikit-learn).

**Naïve Bayes.**   Naïve Bayes methods assume independence between every pair of features allowing the use of Bayes' theorem to find the most probable class given the training data. They are fast to train and very robust due to their simplifying assumptions. We consider three variants of Naïve Bayes: In Gaussian Naïve Bayes the likelihood of the features is assumed to be Gaussian. Multinomial Naïve Bayes is a variant suitable for multinomially distributed data. Bernoulli Naïve Bayes assumes a Bernoulli distribution. However, as shown in our experiments they proved too simplistic of a classifier choice to be effective in the AutoML setting.

**Decision trees.**   Decision trees are one of the most frequently used baseline classifiers in operation. They also constitute the building block of ensemble methods such as random forests which often show strong empirical performance. Basic decision trees (as used in our pipeline) are constructed by recursively splitting the training data into subsets based on the feature values. The criteria for determining the best rule for splitting in scikit-learn are based on a cross-entropy measurement or Gini impurity.

**Ensemble methods.**   The final set of machine learning models we consider are simple, yet powerful, ensemble methods. Concretely we consider AdaBoost, gradient boosting, random forests and extremely randomized trees. Among these, AdaBoost is perhaps the most prototypical ensemble method which combines a sequence of "weak learners" into a weighted majority vote. Successive weak learners are trained with reweighted versions of the training data, where higher weights are assigned to misclassified samples. We use decision trees with a maximum depth of 10 as weak learners. The other ensemble methods we consider also use decision trees as base classifiers: Gradient boosting generalizes the idea of AdaBoost to arbitrarily differentiable loss functions. Random forests and extremely randomized trees are ensembles of decision trees that are trained with a bootstrap sample of the training data. In random forests the best splitting rule is determined by optimizing Gini impurity or information gain among a random subset of the features. Extremely randomized trees use randomly generated splitting rules as candidates and choose the best one.

## A.2   Feature preprocessing algorithms

In addition to the classifier choices AUTO-SKLEARN contains a large set of different feature preprocessing algorithms; which can optionally be selected by the Bayesian optimization algorithm. These again can be separated roughly into 8 categories.

**Matrix decomposition.**   The first category of feature preprocessing methods decomposes the given data into maximally descriptive components. Among these we consider Principal component analysis (PCA), a truncated SVD, Kernel PCA and Independent component analysis (ICA). Principal component analysis (PCA) is perhaps the most well known feature preprocessing method and performs a linear mapping of the data onto its principal components. Truncated SVD is an approximation to PCA which also works in a spare data regime. Kernel PCA is performing principal component analysis in a reproducing kernel Hilbert space, allowing for non-linear mappings. Independent component analysis (ICA) finds basis vectors such that data projected onto these basis have maximum statistical independence.

**Univariate feature selection.**   A second category of feature preprocessing methods which, although simple, often performs well is to "simply" select features based on univariate statistical tests on the dataset. From these, scikit-learn includes: (1) feature selection according to a percentile of the highest scores given some scoring function (such as the feature variance) (2) discarding features lower than a given threshold on a scoring function (this is called *select rates* in the main paper).

**Classification-based feature selection.**   Feature selection can also be performed by more elaborate machine learning methods. We include classification-based feature selection which consists of fitting a classifier to the data and choosing features that the classifier deems to be important for correct classification. Concretely, we allow for the use of $l_1$-regularized linear SVMs for feature selection by fitting the SVM to the data and choosing features corresponding to non-zero model coefficients. Additionally, extremely randomized trees can be used as a preprocessor for feature selection. The relative importance of a feature is calculated as the reduction of the splitting criterion brought by that feature. Then only the most important features are selected.

**Feature clustering.** Instead of feature selection one can also merge features (i.e. add them together) which highly correlate. For this purpose the feature agglomeration preprocessing, implemented in scikit-learn is included in our AutoML system.

**Kernel approximations.** Can be used to approximate kernel functions (such as the RBF kernel) over the dataset without the need for actual (costly) computation of the kernel between all data points. From these we consider *random kitchen sinks* and *nystroem sampling*. Random kitchen sinks map the data to a higher dimensional feature space through a randomized feature map that guarantees that inner product between pairs of points in feature space approximates the evaluation of a kernel (in our case the Gaussian kernel). Nystroem sampling is a technique that accomplishes the same goal by projecting examples on a random subset of the data.

**Polynomial feature expansion.** Simply expands the set of available features by calculating all polynomial combinations (up to a given degree) of the features.

**Feature embeddings.** Project the set of features into a feature space through a non-linear embedding. While there exists a multitude of such embedding methods we consider only embedding through random forests. More precisely our random trees embedding uses an ensemble of totally random trees for unsupervised transformation of the data to a sparse representation. Points are encoded according to the leaf of each tree they are sorted in.

**Sparse representation transformation.** For completeness we also include a simple sparse to dense transformation in our preprocessing pipeline which, while costly, allows us to use algorithms on sparse data that cannot natively handle sparsely represented inputs.

### A.3 Data preprocessing algorithms

Prior to doing feature preprocessing and classification, the data is preprocessed by the following algorithms in the presented order:

1. One Hot Encoding replaces categorical features $f$ with domain $v_1, \ldots, v_k$ by $k$ binary variables, only the $i$-th of which is set to true for data points where $f$ is set to $v_i$.
2. Imputation will replace missing values by the mean, median or most frequent value.
3. Rescaling either standardizes the features to have zero mean and unit variance or rescales them into the range $[0, 1]$. Alternatively, it can normalize samples to have unit length or leave features unscaled.
4. Balancing activates a class weight mechanism of the classification algorithm if it supports one.

## B  Details of AUTO-SKLEARN

As with every robust real-world system, we had to handle many important details in AUTO-SKLEARN. To make the most of our computational power and not get stuck in a very slow run of a certain combination of preprocessing and machine learning algorithm, we implemented measures to prevent such long runs. First, we limited the time for each evaluation of an instantiation of the ML framework, typically to $\frac{1}{10}$ of the overall time limit. We also limited the memory of such evaluations to prevent the operating system from swapping. When an evaluation went over one of those limits, we killed it and returned the worst possible score for the given evaluation metric. For some of the models we employed an iterative training procedure; we instrumented these to still return a performance value when a limit was reached. To further reduce the amount of overly long runs, we forbade several combinations of preprocessors and classification methods: in particular, kernel approximation was forbidden to be active in conjunction with non-linear and tree-based methods as well as the KNN algorithm. (SMAC handles such forbidden combinations natively.) For the same reason we also left out feature learning algorithms, such as dictionary learning.

Another issue in hyperparameter optimization is overfitting and data resampling. Here we had to trade off between running a more robust cross-validation (which comes at little additional overhead in SMAC) and evaluating models on all cross-validation folds to allow for ensemble construction with these models. Thus, for tasks with a rigid time limit of 1h, we used a simple train/test split. In

contrast, we are able to employ ten-fold crossvalidation in our 24h and 30h runs, as well as in our experiments for the human track of the AutoML challenge.

Finally, not every supervised learning task (for example classification with multiple targets), can be solved by all of the algorithms available in AUTO-SKLEARN. Thus, given a new dataset, AUTO-SKLEARN preselects the methods that are suitable for the dataset's properties. Since scikit-learn methods are restricted to numerical input values, we transformed data by applying a one-hot encoding to categorical features. In order to keep the number of dummy features low, we configured a percentage threshold. A value occurring more seldom than this percentage was transformed to a special *other* variable [4].

## C Meta-features

| Meta-feature | Value | | | Calculation time (s) | | |
| --- | --- | --- | --- | --- | --- | --- |
| | Minimum | Mean | Maximum | Minimum | Mean | Maximum |
| class-entropy | 0.64 | 1.92 | 4.70 | 0.00 | 0.00 | 0.00 |
| class-probability-max | 0.04 | 0.43 | 0.90 | 0.00 | 0.00 | 0.00 |
| class-probability-mean | 0.04 | 0.28 | 0.50 | 0.00 | 0.00 | 0.00 |
| class-probability-min | 0.00 | 0.19 | 0.48 | 0.00 | 0.00 | 0.00 |
| class-probability-std | 0.00 | 0.10 | 0.35 | 0.00 | 0.00 | 0.00 |
| dataset-ratio | 0.00 | 0.06 | 0.62 | 0.00 | 0.00 | 0.00 |
| inverse-dataset-ratio | 1.62 | 141.90 | 1620.00 | 0.00 | 0.00 | 0.00 |
| kurtosis-max | -1.30 | 193.43 | 4812.49 | 0.00 | 0.01 | 0.05 |
| kurtosis-mean | -1.30 | 24.32 | 652.23 | 0.00 | 0.01 | 0.05 |
| kurtosis-min | -3.00 | -0.59 | 5.25 | 0.00 | 0.01 | 0.05 |
| kurtosis-std | 0.00 | 48.83 | 1402.86 | 0.00 | 0.01 | 0.05 |
| landmark-1NN* | 0.20 | 0.79 | 1.00 | 0.01 | 0.61 | 8.97 |
| landmark-decision-node-learner* | 0.07 | 0.55 | 0.96 | 0.00 | 0.13 | 1.34 |
| landmark-decision-tree* | 0.20 | 0.78 | 1.00 | 0.00 | 0.49 | 5.23 |
| landmark-lda* | 0.26 | 0.79 | 1.00 | 0.00 | 1.39 | 70.08 |
| landmark-naive-bayes* | 0.10 | 0.68 | 0.97 | 0.00 | 0.06 | 1.05 |
| landmark-random-node-learner* | 0.07 | 0.47 | 0.91 | 0.00 | 0.02 | 0.26 |
| log-dataset-ratio | -7.39 | -3.80 | -0.48 | 0.00 | 0.00 | 0.00 |
| log-inverse-dataset-ratio | 0.48 | 3.80 | 7.39 | 0.00 | 0.00 | 0.00 |
| log-number-of-features | 1.10 | 2.92 | 5.63 | 0.00 | 0.00 | 0.00 |
| log-number-of-instances | 4.04 | 6.72 | 9.90 | 0.00 | 0.00 | 0.00 |
| number-of-Instances-with-missing-values | 0.00 | 96.00 | 2480.00 | 0.00 | 0.00 | 0.01 |
| number-of-categorical-features | 0.00 | 13.25 | 240.00 | 0.00 | 0.00 | 0.00 |
| number-of-classes | 2.00 | 6.58 | 28.00 | 0.00 | 0.00 | 0.00 |
| number-of-features | 3.00 | 33.91 | 279.00 | 0.00 | 0.00 | 0.00 |
| number-of-features-with-missing-values | 0.00 | 3.54 | 34.00 | 0.00 | 0.00 | 0.00 |
| number-of-instances | 57.00 | 2126.33 | 20000.00 | 0.00 | 0.00 | 0.00 |
| number-of-missing-values | 0.00 | 549.49 | 22175.00 | 0.00 | 0.00 | 0.00 |
| number-of-numeric-features | 0.00 | 20.67 | 216.00 | 0.00 | 0.00 | 0.00 |
| pca-95percent* | 0.02 | 0.52 | 1.00 | 0.00 | 0.00 | 0.00 |
| pca-kurtosis-first-pc* | -2.00 | 13.38 | 730.92 | 0.00 | 0.00 | 0.01 |
| pca-skewness-first-pc* | -27.07 | -0.16 | 6.46 | 0.00 | 0.00 | 0.04 |
| percentage-of-Instances-with-missing-values | 0.00 | 0.14 | 1.00 | 0.00 | 0.00 | 0.00 |
| percentage-of-features-with-missing-values | 0.00 | 0.16 | 1.00 | 0.00 | 0.00 | 0.00 |
| percentage-of-missing-values | 0.00 | 0.03 | 0.65 | 0.00 | 0.00 | 0.00 |
| ratio-categorical-to-numerical | 0.00 | 1.35 | 33.00 | 0.00 | 0.00 | 0.00 |
| ratio-numerical-to-categorical | 0.00 | 0.49 | 7.00 | 0.00 | 0.00 | 0.00 |
| skewness-max | 0.00 | 5.34 | 67.41 | 0.00 | 0.00 | 0.04 |
| skewness-mean | -0.56 | 1.27 | 14.71 | 0.00 | 0.00 | 0.04 |
| skewness-min | -21.19 | -0.62 | 1.59 | 0.00 | 0.00 | 0.04 |
| skewness-std | 0.00 | 1.60 | 18.89 | 0.00 | 0.01 | 0.05 |
| symbols-max | 0.00 | 13.09 | 429.00 | 0.00 | 0.00 | 0.00 |
| symbols-mean | 0.00 | 3.01 | 41.38 | 0.00 | 0.00 | 0.00 |
| symbols-min | 0.00 | 1.44 | 12.00 | 0.00 | 0.00 | 0.00 |
| symbols-std | 0.00 | 3.06 | 107.21 | 0.00 | 0.00 | 0.00 |
| symbols-sum | 0.00 | 71.04 | 1648.00 | 0.00 | 0.00 | 0.00 |

Table 1: List of implemented meta-features. Meta-features marked with an asterisks were only used to do the dataset clustering in Section 6

## D Ensemble selection

Pseudocode explaining our implementation of the ensemble selection algorithm [5].

---
**Procedure 1:** EnsembleSelection($M, S$)

---
**Input** : Models $M$, Ensemble size $S$ , $n = |M|$
**Output** : Ensemble $E$

1 $E \leftarrow \emptyset$
2 **for** $i = 0 \ldots S$ **do**
3      $b \leftarrow \text{argmax}_{j=0 \ldots n} performance(E \cup M[j])$
4      $E \leftarrow E \cup M[b]$
5 **return** $E$

---

# E   Configuration spaces for different dataset properties

### (a) classifiers

| name | #λ | cat (cond) | cont (cond) |
|---|---|---|---|
| AdaBoost (AB) | 4 | 1 (-) | 3 (-) |
| Bernoulli naïve Bayes | 2 | 1 (-) | 1 (-) |
| decision tree (DT) | 4 | 1 (-) | 3 (-) |
| extreml. rand. trees | 5 | 2 (-) | 3 (-) |
| Gaussian naïve Bayes | - | - | - |
| gradient boosting (GB) | 6 | - | 6 (-) |
| kNN | 3 | 2 (-) | 1 (-) |
| LDA | 4 | 1 (-) | 3 (1) |
| linear SVM | 4 | 2 (-) | 2 (-) |
| kernel SVM | 7 | 2 (-) | 5 (2) |
| multinomial naïve Bayes | 2 | 1 (-) | 1 (-) |
| passive aggressive | 3 | 1 (-) | 2 (-) |
| QDA | 2 | - | 2 (-) |
| random forest (RF) | 5 | 2 (-) | 3 (-) |
| SGD | 10 | 4 (-) | 6 (3) |

### (b) preprocessing methods

| name | #λ | cat (cond) | cont (cond) |
|---|---|---|---|
| densifier | - | - | - |
| extreml. rand. trees prepr. | 5 | 2 (-) | 3 (-) |
| kernel PCA | 5 | 1 (-) | 4 (3) |
| rand. kitchen sinks | 2 | - | 2 (-) |
| linear SVM prepr. | 3 | 1 (-) | 2 (-) |
| no preprocessing | - | - | - |
| nystroem sampler | 5 | 1 (-) | 4 (3) |
| random trees embed. | 4 | - | 4 (-) |
| select percentile | 2 | 1 (-) | 1 (-) |
| select rates | 3 | 2 (-) | 1 (-) |
| truncated SVD | 1 | - | 1 (-) |
| one-out-of-k encoding | 2 | 1 (-) | 1 (1) |
| imputation | 1 | 1 (-) | - |
| balancing | 1 | 1 (-) | - |
| rescaling | 1 | 1 (-) | - |

Table 2: Number of hyperparameters for each possible classifier (left) and feature preprocessing method (right) for a **binary classification** dataset in **sparse** representation.

### (a) classifiers

| name | #λ | cat (cond) | cont (cond) |
|---|---|---|---|
| AdaBoost (AB) | 4 | 1 (-) | 3 (-) |
| decision tree (DT) | 4 | 1 (-) | 3 (-) |
| extreml. rand. trees | 5 | 2 (-) | 3 (-) |
| Gaussian naïve Bayes | - | - | - |
| gradient boosting (GB) | 6 | - | 6 (-) |
| kNN | 3 | 2 (-) | 1 (-) |
| LDA | 4 | 1 (-) | 3 (1) |
| linear SVM | 4 | 2 (-) | 2 (-) |
| kernel SVM | 7 | 2 (-) | 5 (2) |
| multinomial naïve Bayes | 2 | 1 (-) | 1 (-) |
| passive aggressive | 3 | 1 (-) | 2 (-) |
| QDA | 2 | - | 2 (-) |
| random forest (RF) | 5 | 2 (-) | 3 (-) |
| SGD | 10 | 4 (-) | 6 (3) |

### (b) preprocessing methods

| name | #λ | cat (cond) | cont (cond) |
|---|---|---|---|
| extreml. rand. trees prepr. | 5 | 2 (-) | 3 (-) |
| fast ICA | 4 | 3 (-) | 1 (1) |
| feature agglomeration | 4 | 3 (-) | 1 (-) |
| kernel PCA | 5 | 1 (-) | 4 (3) |
| rand. kitchen sinks | 2 | - | 2 (-) |
| linear SVM prepr. | 3 | 1 (-) | 2 (-) |
| no preprocessing | - | - | - |
| nystroem sampler | 5 | 1 (-) | 4 (3) |
| PCA | 2 | 1 (-) | 1 (-) |
| polynomial | 3 | 2 (-) | 1 (-) |
| random trees embed. | 4 | - | 4 (-) |
| select percentile | 2 | 1 (-) | 1 (-) |
| select rates | 3 | 2 (-) | 1 (-) |
| one-out-of-k encoding | 2 | 1 (-) | 1 (1) |
| imputation | 1 | 1 (-) | - |
| balancing | 1 | 1 (-) | - |
| rescaling | 1 | 1 (-) | - |

Table 3: Number of hyperparameters for each possible classifier (left) and feature preprocessing method (right) for a **multiclass classification** dataset in **dense** representation.

### (a) classifiers

| name | #λ | cat (cond) | cont (cond) |
|---|---|---|---|
| AdaBoost (AB) | 4 | 1 (-) | 3 (-) |
| decision tree (DT) | 4 | 1 (-) | 3 (-) |
| extreml. rand. trees | 5 | 2 (-) | 3 (-) |
| Gaussian naïve Bayes | - | - | - |
| gradient boosting (GB) | 6 | - | 6 (-) |
| kNN | 3 | 2 (-) | 1 (-) |
| LDA | 4 | 1 (-) | 3 (1) |
| linear SVM | 4 | 2 (-) | 2 (-) |
| kernel SVM | 7 | 2 (-) | 5 (2) |
| multinomial naïve Bayes | 2 | 1 (-) | 1 (-) |
| passive aggressive | 3 | 1 (-) | 2 (-) |
| QDA | 2 | - | 2 (-) |
| random forest (RF) | 5 | 2 (-) | 3 (-) |
| SGD | 10 | 4 (-) | 6 (3) |

### (b) preprocessing methods

| name | #λ | cat (cond) | cont (cond) |
|---|---|---|---|
| densifier | - | - | - |
| extreml. rand. trees prepr. | 5 | 2 (-) | 3 (-) |
| kernel PCA | 5 | 1 (-) | 4 (3) |
| rand. kitchen sinks | 2 | - | 2 (-) |
| linear SVM prepr. | 3 | 1 (-) | 2 (-) |
| no preprocessing | - | - | - |
| nystroem sampler | 5 | 1 (-) | 4 (3) |
| random trees embed. | 4 | - | 4 (-) |
| select percentile | 2 | 1 (-) | 1 (-) |
| select rates | 3 | 2 (-) | 1 (-) |
| truncated SVD | 1 | - | 1 (-) |
| one-out-of-k encoding | 2 | 1 (-) | 1 (1) |
| imputation | 1 | 1 (-) | - |
| balancing | 1 | 1 (-) | - |
| rescaling | 1 | 1 (-) | - |

Table 4: Number of hyperparameters for each possible classifier (left) and feature preprocessing method (right) for a **multiclass classification** dataset in **sparse** representation.

# F Properties of datasets used in the experiments

| Name | #Continuous | #Nominal | #Classes | Sparse? | Missing Values | #Training Samples | #Test Samples |
|------|-------------|----------|----------|---------|----------------|-------------------|---------------|
| Abalone | 7 | 1 | 26 | - | - | 2924 | 1253 |
| Amazon | 10000 | 0 | 50 | - | - | 1050 | 450 |
| Car | 0 | 6 | 4 | - | - | 1210 | 518 |
| Cifar10 | 3072 | 0 | 10 | - | - | 50000 | 10000 |
| Cifar-10-Small | 3072 | 0 | 10 | - | - | 10000 | 10000 |
| Convex | 784 | 0 | 2 | - | - | 8000 | 50000 |
| Dexter | 20000 | 0 | 2 | X | - | 420 | 180 |
| Dorothea | 100000 | 0 | 2 | X | - | 805 | 345 |
| GermanCredit | 7 | 13 | 2 | - | - | 700 | 300 |
| Gisette | 5000 | 0 | 2 | - | - | 4900 | 2100 |
| KDD09-Appetency | 192 | 38 | 2 | - | X | 35000 | 15000 |
| KR-vs-KP | 0 | 36 | 2 | - | - | 2238 | 958 |
| Madelon | 500 | 0 | 2 | - | - | 1820 | 780 |
| MNIST Basic | 784 | 0 | 10 | - | - | 12000 | 50000 |
| Rot. MNIST + BI | 784 | 0 | 10 | - | - | 12000 | 50000 |
| Secom | 590 | 0 | 2 | - | X | 1097 | 470 |
| Semeion | 256 | 0 | 10 | - | - | 1116 | 477 |
| Shuttle | 9 | 0 | 7 | - | - | 43500 | 14500 |
| Waveform | 40 | 0 | 3 | - | - | 3500 | 1500 |
| Wine Quality | 11 | 0 | 7 | - | - | 3429 | 1469 |
| Yeast | 8 | 0 | 10 | - | - | 1039 | 445 |

Table 5: Auto-WEKA datasets [6].

| ID | Name | #Continuous | #Nominal | #Classes | Sparse? | Missing Values | #Training Samples | #Test Samples |
|----|------|-------------|----------|----------|---------|----------------|-------------------|---------------|
| 38 | Sick | 7 | 22 | 2 | - | X | 2527 | 1245 |
| 46 | Splice | 0 | 60 | 3 | - | - | 2137 | 1053 |
| 179 | adult | 2 | 12 | 2 | - | X | 32724 | 16118 |
| 184 | KROPT | 0 | 6 | 18 | - | - | 18797 | 9259 |
| 554 | MNIST | 784 | 0 | 10 | - | - | 46900 | 23100 |
| 772 | quake | 3 | 0 | 2 | - | - | 1459 | 719 |
| 917 | fri_c1_1000_25 (binarized) | 25 | 0 | 2 | - | - | 670 | 330 |
| 1049 | pc4 | 37 | 0 | 2 | - | - | 976 | 482 |
| 1111 | KDDCup09 Appetency | 192 | 38 | 2 | (X) | X | 33500 | 16500 |
| 1120 | Magic Telescope | 10 | 0 | 2 | - | - | 12743 | 6277 |
| 1128 | OVA Breast | 10935 | 0 | 2 | - | - | 1035 | 510 |
| 293 | Covertype (binarized) | 54 | 0 | 2 | X | - | 389278 | 191734 |
| 389 | fbis_wc | 2000 | 0 | 17 | X | - | 1651 | 812 |

Table 6: Representative datasets for the 13 clusters obtained via g-means clustering of the 140 datasets' meta-feature vectors.

| ID | Name | #Continuous | #Nominal | #Classes | Sparse? | Missing Values | #Training Samples | #Test Samples |
|---|---|---|---|---|---|---|---|---|
| 3 | kr-vs-kp | 0 | 36 | 2 | - | - | 2141 | 1055 |
| 6 | letter | 16 | 0 | 26 | - | - | 13402 | 6598 |
| 12 | mfeat-factors | 216 | 0 | 10 | - | - | 1340 | 660 |
| 14 | mfeat-fourier | 76 | 0 | 10 | - | - | 1340 | 660 |
| 16 | mfeat-karhunen | 64 | 0 | 10 | - | - | 1340 | 660 |
| 18 | mfeat-morphological | 6 | 0 | 10 | - | - | 1340 | 660 |
| 21 | car | 0 | 6 | 4 | - | - | 1157 | 571 |
| 22 | mfeat-zernike | 47 | 0 | 10 | - | - | 1340 | 660 |
| 23 | cmc | 2 | 7 | 3 | - | - | 986 | 487 |
| 24 | mushroom | 0 | 22 | 2 | - | X | 5443 | 2681 |
| 26 | nursery | 0 | 8 | 5 | - | - | 8682 | 4278 |
| 28 | optdigits | 64 | 0 | 10 | - | - | 3765 | 1855 |
| 30 | page-blocks | 10 | 0 | 5 | - | - | 3666 | 1807 |
| 31 | credit-g | 7 | 13 | 2 | - | - | 670 | 330 |
| 32 | pendigits | 16 | 0 | 10 | - | - | 7364 | 3628 |
| 36 | segment | 19 | 0 | 7 | - | - | 1547 | 763 |
| 38 | sick | 7 | 22 | 2 | - | X | 2527 | 1245 |
| 44 | spambase | 57 | 0 | 2 | - | - | 3082 | 1519 |
| 46 | splice | 0 | 60 | 3 | - | - | 2137 | 1053 |
| 57 | hypothyroid | 7 | 22 | 4 | - | X | 2527 | 1245 |
| 60 | waveform-5000 | 40 | 0 | 3 | - | - | 3351 | 1649 |
| 179 | adult | 2 | 12 | 2 | - | X | 32724 | 16118 |
| 180 | covertype | 14 | 40 | 7 | - | - | 73962 | 36431 |
| 181 | yeast | 8 | 0 | 10 | - | - | 991 | 493 |
| 182 | satimage | 36 | 0 | 6 | - | - | 4308 | 2122 |
| 184 | kropt | 0 | 6 | 18 | - | - | 18797 | 9259 |
| 185 | baseball | 15 | 1 | 3 | - | X | 897 | 443 |
| 273 | IMDB.drama | 1001 | 0 | 2 | X | - | 81007 | 39899 |
| 293 | covertype | 54 | 0 | 2 | X | - | 389278 | 191734 |
| 300 | isolet | 617 | 0 | 26 | - | - | 5224 | 2573 |
| 351 | codrna | 8 | 0 | 2 | X | - | 327338 | 161227 |
| 354 | poker | 10 | 0 | 2 | X | - | 686756 | 338254 |
| 357 | vehicle_sensIT | 100 | 0 | 2 | X | - | 66012 | 32516 |
| 389 | fbis.wc | 2000 | 0 | 17 | X | - | 1651 | 812 |
| 390 | new3s.wc | 26832 | 0 | 44 | X | - | 6401 | 3157 |
| 391 | re0.wc | 2886 | 0 | 13 | X | - | 1007 | 497 |
| 392 | oh0.wc | 3182 | 0 | 10 | X | - | 672 | 331 |
| 393 | la2s.wc | 12432 | 0 | 6 | X | - | 2059 | 1016 |
| 395 | re1.wc | 3758 | 0 | 25 | X | - | 1109 | 548 |
| 396 | la1s.wc | 13195 | 0 | 6 | X | - | 2146 | 1058 |
| 398 | wap.wc | 8460 | 0 | 20 | X | - | 1044 | 516 |
| 399 | ohscal.wc | 11465 | 0 | 10 | X | - | 7478 | 3684 |
| 401 | oh10.wc | 3238 | 0 | 10 | X | - | 702 | 348 |
| 554 | mnist_784 | 784 | 0 | 10 | - | - | 46900 | 23100 |
| 679 | rmftsa_sleepdata | 2 | 0 | 4 | - | - | 687 | 337 |
| 715 | fri_c3_1000_25 | 25 | 0 | 2 | - | - | 670 | 330 |
| 718 | fri_c4_1000_100 | 100 | 0 | 2 | - | - | 670 | 330 |
| 720 | abalone | 7 | 1 | 2 | - | - | 2798 | 1379 |
| 722 | pol | 48 | 0 | 2 | - | - | 10050 | 4950 |
| 723 | fri_c4_1000_25 | 25 | 0 | 2 | - | - | 670 | 330 |
| 727 | 2dplanes | 10 | 0 | 2 | - | - | 27314 | 13454 |
| 728 | analcatdata_supreme | 7 | 0 | 2 | - | - | 2714 | 1338 |
| 734 | ailerons | 40 | 0 | 2 | - | - | 9212 | 4538 |
| 735 | cpu_small | 12 | 0 | 2 | - | - | 5488 | 2704 |
| 737 | space_ga | 6 | 0 | 2 | - | - | 2081 | 1026 |
| 740 | fri_c3_1000_10 | 10 | 0 | 2 | - | - | 670 | 330 |
| 741 | rmftsa_sleepdata | 1 | 1 | 2 | - | - | 686 | 338 |
| 743 | fri_c1_1000_5 | 5 | 0 | 2 | - | - | 670 | 330 |
| 751 | fri_c4_1000_10 | 10 | 0 | 2 | - | - | 670 | 330 |
| 752 | puma32H | 32 | 0 | 2 | - | - | 5488 | 2704 |
| 761 | cpu_act | 21 | 0 | 2 | - | - | 5488 | 2704 |
| 772 | quake | 3 | 0 | 2 | - | - | 1459 | 719 |
| 797 | fri_c4_1000_50 | 50 | 0 | 2 | - | - | 670 | 330 |
| 799 | fri_c0_1000_5 | 5 | 0 | 2 | - | - | 670 | 330 |
| 803 | delta_ailerons | 5 | 0 | 2 | - | - | 4776 | 2353 |
| 806 | fri_c3_1000_50 | 50 | 0 | 2 | - | - | 670 | 330 |
| 807 | kin8nm | 8 | 0 | 2 | - | - | 5488 | 2704 |
| 813 | fri_c3_1000_5 | 5 | 0 | 2 | - | - | 670 | 330 |
| 816 | puma8NH | 8 | 0 | 2 | - | - | 5488 | 2704 |
| 819 | delta_elevators | 6 | 0 | 2 | - | - | 6376 | 3141 |
| 821 | house_16H | 16 | 0 | 2 | - | - | 15265 | 7519 |
| 822 | cal_housing | 8 | 0 | 2 | - | - | 13828 | 6812 |
| 823 | houses | 8 | 0 | 2 | - | - | 13828 | 6812 |
| 833 | bank32nh | 32 | 0 | 2 | - | - | 5488 | 2704 |
| 837 | fri_c1_1000_50 | 50 | 0 | 2 | - | - | 670 | 330 |
| 843 | house_8L | 8 | 0 | 2 | - | - | 15265 | 7519 |
| 845 | fri_c0_1000_10 | 10 | 0 | 2 | - | - | 670 | 330 |
| 846 | elevators | 18 | 0 | 2 | - | - | 11121 | 5478 |
| 847 | wind | 14 | 0 | 2 | - | - | 4404 | 2170 |

Table 7: All datasets which were used for generating metadata and the experiments in Section 5 of the main paper.

| ID | Name | #Continuous | #Nominal | #Classes | Sparse? | Missing Values | #Training Samples | #Test Samples |
|----|------|-------------|----------|----------|---------|----------------|-------------------|---------------|
| 849 | fri_c0_1000_25 | 25 | 0 | 2 | - | - | 670 | 330 |
| 866 | fri_c2_1000_50 | 50 | 0 | 2 | - | - | 670 | 330 |
| 871 | pollen | 5 | 0 | 2 | - | - | 2578 | 1270 |
| 881 | mv | 7 | 3 | 2 | - | - | 27314 | 13454 |
| 897 | colleges_aaup | 13 | 2 | 2 | - | X | 777 | 384 |
| 901 | fried | 10 | 0 | 2 | - | - | 27314 | 13454 |
| 903 | fri_c2_1000_25 | 25 | 0 | 2 | - | - | 670 | 330 |
| 904 | fri_c0_1000_50 | 50 | 0 | 2 | - | - | 670 | 330 |
| 910 | fri_c1_1000_10 | 10 | 0 | 2 | - | - | 670 | 330 |
| 912 | fri_c2_1000_5 | 5 | 0 | 2 | - | - | 670 | 330 |
| 913 | fri_c2_1000_10 | 10 | 0 | 2 | - | - | 670 | 330 |
| 914 | balloon | 1 | 0 | 2 | - | - | 1340 | 661 |
| 917 | fri_c1_1000_25 | 25 | 0 | 2 | - | - | 670 | 330 |
| 923 | visualizing_soil | 3 | 1 | 2 | - | - | 5789 | 2852 |
| 930 | colleges_usnews | 32 | 1 | 2 | - | X | 872 | 430 |
| 934 | socmob | 1 | 4 | 2 | - | - | 774 | 382 |
| 953 | splice | 0 | 60 | 2 | - | - | 2137 | 1053 |
| 958 | segment | 19 | 0 | 2 | - | - | 1547 | 763 |
| 959 | nursery | 0 | 8 | 2 | - | - | 8683 | 4277 |
| 962 | mfeat-morphological | 6 | 0 | 2 | - | - | 1340 | 660 |
| 966 | analcatdata_halloffame | 15 | 1 | 2 | - | X | 897 | 443 |
| 971 | mfeat-fourier | 76 | 0 | 2 | - | - | 1340 | 660 |
| 976 | kdd_JapaneseVowels | 14 | 0 | 2 | - | - | 6673 | 3288 |
| 977 | letter | 16 | 0 | 2 | - | - | 13400 | 6600 |
| 978 | mfeat-factors | 216 | 0 | 2 | - | - | 1340 | 660 |
| 979 | waveform-5000 | 40 | 0 | 2 | - | - | 3350 | 1650 |
| 980 | optdigits | 64 | 0 | 2 | - | - | 3765 | 1855 |
| 991 | car | 0 | 6 | 2 | - | - | 1157 | 571 |
| 993 | kdd_ipums_la_97-small | 33 | 27 | 2 | - | X | 4702 | 2317 |
| 995 | mfeat-zernike | 47 | 0 | 2 | - | - | 1340 | 660 |
| 100 | hypothyroid | 7 | 22 | 2 | - | X | 2527 | 1245 |
| 100 | kdd_ipums_la_98-small | 16 | 39 | 2 | - | X | 5014 | 2471 |
| 101 | kdd_ipums_la_99-small | 15 | 41 | 2 | - | X | 5925 | 2919 |
| 101 | pendigits | 16 | 0 | 2 | - | - | 7364 | 3628 |
| 102 | mfeat-karhunen | 64 | 0 | 2 | - | - | 1340 | 660 |
| 102 | page-blocks | 10 | 0 | 2 | - | - | 3666 | 1807 |
| 103 | sylva_agnostic | 216 | 0 | 2 | - | - | 9644 | 4751 |
| 104 | sylva_prior | 108 | 0 | 2 | - | - | 9644 | 4751 |
| 104 | gina_prior2 | 784 | 0 | 10 | - | - | 2322 | 1146 |
| 104 | pc4 | 37 | 0 | 2 | - | - | 976 | 482 |
| 105 | pc3 | 37 | 0 | 2 | - | - | 1047 | 516 |
| 105 | jm1 | 21 | 0 | 2 | - | X | 7292 | 3593 |
| 105 | mc1 | 38 | 0 | 2 | - | - | 6342 | 3124 |
| 106 | kc1 | 21 | 0 | 2 | - | - | 1413 | 696 |
| 106 | pc1 | 21 | 0 | 2 | - | - | 743 | 366 |
| 106 | pc2 | 36 | 0 | 2 | - | - | 3744 | 1845 |
| 1111 | KDDCup09_appetency | 192 | 38 | 2 | - | X | 33500 | 16500 |
| 1112 | KDDCup09_churn | 192 | 38 | 2 | - | X | 33500 | 16500 |
| 1114 | KDDCup09_upselling | 192 | 38 | 2 | - | X | 33500 | 16500 |
| 1116 | musk | 166 | 1 | 2 | - | - | 4420 | 2178 |
| 1119 | adult-census | 6 | 8 | 2 | - | X | 21815 | 10746 |
| 1120 | MagicTelescope | 10 | 0 | 2 | - | - | 12743 | 6277 |
| 1128 | OVA_Breast | 10935 | 0 | 2 | - | - | 1035 | 510 |
| 1130 | OVA_Lung | 10935 | 0 | 2 | - | - | 1035 | 510 |
| 1134 | OVA_Kidney | 10935 | 0 | 2 | - | - | 1035 | 510 |
| 1138 | OVA_Uterus | 10935 | 0 | 2 | - | - | 1035 | 510 |
| 1139 | OVA_Omentum | 10935 | 0 | 2 | - | - | 1035 | 510 |
| 1142 | OVA_Endometrium | 10935 | 0 | 2 | - | - | 1035 | 510 |
| 1146 | OVA_Prostate | 10935 | 0 | 2 | - | - | 1035 | 510 |
| 1161 | OVA_Colon | 10935 | 0 | 2 | - | - | 1035 | 510 |
| 1166 | OVA_Ovary | 10935 | 0 | 2 | - | - | 1035 | 510 |

Table 8: All datasets which were used for generating metadata and the experiments in Section 5 of the main paper (continued).

# G   Setup section 4

To compare against Auto-WEKA, we used 21 datasets (detailed in Table 5) with their original train/test split [6], a walltime limit of 30 hours, 10-fold cross validation (where the evaluation of each fold was allowed to take 150 minutes, except for hyperopt-sklearn which uses a 80/20 train/test split), and 10 independent optimization runs with SMAC on each dataset. Our results for Auto-WEKA resemble those of Thornton et al. [7], with minor differences caused by our faster machines: All our experiments ran on Intel Xeon E5-2650 v2 eight-core processors with 2.60GHz and 4GiB of RAM. We allowed the machine learning framework to use 3GiB and reserved the rest for SMAC. All experiments used Auto-WEKA 0.5 and scikit-learn 0.16.1.

# H   Evaluation of our new AutoML methods - additional plot

Figure 1: Average rank of all four AUTO-SKLEARN versions ranked by balanced error (BER) across 140 datasets. In contrast to the plot in the main paper, this plot is on a log-scale. Due to the little additional overhead that meta-learning and ensemble selection cause, vanilla AUTO-SKLEARN is able to achieve the best rank within the first 10 seconds as it produces predictions before the other AUTO-SKLEARN variants finish training their first model.

# I Average rank over datasets for optimizing single classifiers and preprocessors compared to AUTO-SKLEARN

Figure 2: Average rank. We compare test performance over time for optimizing each classifier with all preprocessing methods separately with optimizing the joint space AUTO-SKLEARN. We optimize each method for one day. Each line shows the average across 13 datasets; for each dataset drew a bootstrap sample of 100 joint runs and computed the average rank across these runs.

Figure 3: Average rank for different preprocessing methods on a dense dataset. We compare test performance over time for optimizing each preprocessing method with all classifier with optimizing the joint space AUTO-SKLEARN. We optimize each method for one day. Each line shows the average across 13 datasets; for each dataset drew a bootstrap sample of 100 joint runs and computed the average rank across these runs.

(a) Classifiers on dataset 554

(b) Classifiers on dataset 1049

Figure 4: Performance (balanced classification error, BER) of different subspaces compared to AUTO-SKLEARN over time. We show the median test performance over time for all classifiers with all preprocessing methods separately with optimizing the joint space.