[Reviews · NeurIPS 2015]

Submitted by Assigned_Reviewer_1

# Summary

The authors introduce a new automated machine learning (autoML) framework, in which the data preprocessing, feature preprocessing, algorithm choice and hyperparameters tuning are all done without human intervention, using bayesian optimization. They bring two contributions to existing autoML methods: a meta-learning component, in which they use a list of past datasets to warm-start the bayesian optimizer, and a ensemble construction component, which reuses the ranking established by the bayesian optimizer to combine the best methods instead of just using the one found by the bayesian optimizer, for more robustness.

They compare their system, auto-sklearn, to an existing autoML system, Auto-WEKA, and find that they outperform it in a majority of cases. They also compare variations of their system without (some of) the two novel components (meta-learning and ensemble construction) and show that the meta-learning component is the most helpful.

# Quality

The evaluation of the new system is very thorough, with systematic comparison with the state of the art (auto-WEKA) on several datasets. All results are clearly reported, with good discussion of the different cases in which their system beats/is beaten by auto-WEKA. The greedy ensemble construction method is not really justified, though, with no comparison of the greedy technique with other existing ensemble methods.

# Clarity

The paper is very well written and clearly organized. The contributions are clearly stated. The evaluation part could be improved, however: the metrics are not explained in details (what is 1-BAC? how do you compute the average ranks, and what is their meaning?), and some results could be discussed better (why is there an intermediary regime in Fig. 3 where the ranks seem more "extreme", and then go back to something more balanced).

# Originality

The authors combined three well-established concepts (autoML, meta-learning warm-start and ensemble methods) into one global system.

# Significance

Designing a robust, efficient and accurate autoML system is an important problem, as it would allow non-experts to easily apply machine learning algorithms to their problem. The system proposed by the authors benefits a lot from meta-learning, which enables it to take past datasets into account. As this list of datasets will continuously grow, their method will keep improving.

# Pros

- Improved performance using meta-learning - Thorough evaluation of many datasets - Based on a well-known framework, code will be published

# Cons

- Performance metrics not well defined - No discussion of the choice of the greedy ensemble algorithm

# Typos

- Definition 2 uses lower case k, but eq. 1 uses capital K, for what I think is the same thing - Undefined table references on page 4 (first lines of the last two paragraphs)
Summary: Promising improvement of automated machine learning with meta-learning, implemented on top of a recognized machine learning framework. The evaluation metrics could be clearer.

Submitted by Assigned_Reviewer_2

This paper proposes a new Auto-ML system based on scikit-learn. They

claim their method outperforms the rest.

I am stuck in the middle with this paper. On the one hand I like their

approach of combining the meta-feature learning as a (warm) starting

point for their bayesian optimization based algorithm selector. Moreover,

the experimental results are quite positive and they have tried to

interpret them which I like. On the other hand however, the presentation

is poor as firstly the distinction of their method is hidden in sections

2.1 and 2.2 rather than being clearly stated upfront. Furthermore no

details are provided about the bayesian optimization method they employ.

Table numbers in places in the text are question marks. It seems to me

that the paper was hurriedly written.
Summary: Overall improved organization and more details about their method would

significantly improve the quality of their exposition.

Submitted by Assigned_Reviewer_3

SUMMARY

The authors introduce a new automated machine learning (autoML) system for classification, as well as associated technical advances that improve upon its baseline performance.

The new system is referred to as auto-sklearn, since it is built upon the scikit-learn library of machine learning algorithms.

The system uses 16 classification algorithms, 11 preprocessing methods, and 3 data preprocessing methods.

The baseline system involves optimizing over algorithms and hyperparameters for each algorithm, to find a choice that minimizes the cross-validation error.

This is accompanied by two further contributions. First, the authors experiment with meta-learning to assist the optimization; this involves extracting meta-features from a large collection of data sets (140, in this case), and characterizing the types of data sets on which each algorithm performs well.

Second, the authors experiment with automated construction of ensembles to further improve the classification performance; this is done via an efficient ensemble selection method combining intermediate points evaluated during the optimization.

The performance of the baseline system, as well as the augmented system with and without meta-learning and ensembling, is evaluated on 140 diverse data sets. The baseline system tends to provide better performance compared to two existing autoML systems: Auto-WEKA and hyperopt-sklearn. On top of this, employing meta-learning significantly improves performance beyond the baseline, and using ensembling in addition improves it still further.

The authors also provide a detailed analysis of the performance of the 16 individual algorithms, providing insight into the relative merits of various standard classifiers.

COMMENTS

This work represents a significant effort in terms of engineering and computation, for which authors are to be commended.

The article is clearly written and easy to read. The motivation, formulation of the problem, description of the proposed method, description of experiments, and presentation of results are all excellent.

The proposed system demonstrates an impressive advance in autoML performance. The use of meta-learning, in particular, is very interesting.

The observation that ensemble selection can be used to efficiently combine intermediate points from the optimization is also nice.

My criticisms are relatively minor.

1. A possible criticism is that this work largely represents an engineering and computational effort, with few significant advances in terms of novel techniques or theory. However, I believe this is outweighed by the scale of the effort, the strength of the performance demonstrated, and the potential utility of this tool.

2. Line 92-93: If the number of algorithms m is large, could this lead to overfitting, even though we are optimizing with respect to cross-validation error?

3. Do the authors have any insights into how best to choose the number and type of algorithms to include in the system? The current system seems to be based simply on what is available in scikit-learn. Perhaps in future work, it would be interesting to know if there is a principled way to choose the composition of algorithms to be included, beyond the general suggestion that it helps to have a diversity of algorithms.

4. The meta-learning approach is appealing. However, no explanation is given for why the authors chose the particular approach that they did. The approach chosen is roughly analogous to k-nearest neighbors, but one can imagine many other possibilities involving other classifiers or regression algorithms. Are there reasons to prefer the chosen approach over other approaches?

5. Perhaps in future work, it would be interesting if the meta-learning aspect of the system could be formalized via a mathematical objective function that could be optimized in order to improve the performance of the overall system.

6. Line 150: Minor point --- I don't get how this is Bayesian... I would think a Bayesian approach would involve something like putting a prior on the algorithms/hyperparameters, and sampling from the posterior.

7. Tables 4 and 5: Minor point --- it might be helpful to move the auto-sklearn column to the front or back, in order to highlight it. Or maybe to sort the columns alphabetically or by aggregate performance.
Summary: This article introduce a new framework for automated machine learning (autoML) based on scikit-learn, using meta-learning to assist the optimization, and ensemble selection to increase performance. The proposed system exhibits improvements in performance over existing autoML systems, and appears to represent a significant advance in autoML technology. I recommend the article for publication.

Submitted by Assigned_Reviewer_4

The authors propose a framework for automated machine learning (AutoML), dubbed auto-sklearn, which aims to automatically train a machine learning method without human supervision on novel datasets. The described approach consists of three elements: data processing, feature processing and classifier choice, giving rise to a set of 132 heterogeneous hyper-parameters that are tuned via the Bayesian optimizer SMAC. The proposed AutoML framework also comprises a component for meta-learning, i.e., by extracting meta-features from datasets, it allows to initialize the framework configuration from an array of the best configurations for the datasets that are similar to the current one. A further element, not described in detail in the main paper, builds an ensemble of classifiers in an on-line fashion, while the configuration is being optimized.

I found the paper highly interesting, clearly written and well illustrated. The empirical evaluation is extensive and shows that auto-sklearn can consistently and automatically learn a good classifier. However, not being very familiar with the AutoML literature, I am unable to precisely judge its originality and significance in context. It would be interesting to assess the robustness of the proposed method for learning a classifier on a novel dataset which is much different from previously seen ones and concurrently to analyse the impact of the meta-learning stage from varying collection of datasets.
Summary: An efficient automatic machine learning framework based on the SMAC Bayesian optimizer is proposed that outperforms auto-WEKA and wins a recent AutoML competition. Many interesting and important details are relegated to the appendix, making me feel that this paper may be better suited for a longer journal version.

Author Feedback
Author rebuttal: We first want to thank all reviewers for their thorough, positive reviews and their insightful remarks.
We are glad that the reviewers liked the strong performance and thorough evaluation of our method, the potential utility of our system and the paper's clarity.
We reply to the reviewers' questions and comments in turn. References we use here refer to the references in the paper.

Assigned_Reviewer_1:

[Comparison of the greedy ensemble construction method]
We agree that we should have discussed our evaluation of the ensemble construction method in more detail. We chose this approach based on experiments using stacking, global weight optimization, simple averaging, and a bagging variant, which turned out to be prone to overfitting. We will add a discussion on these experiments to the paper.

[Explanation of the used evaluation metric]
We agree that we should explain our evaluation metric more clearly. In brief: Balanced Accuracy (BAC) is the average of class-wise accuracy for classification problems; it is, e.g., used in the AutoML challenge [1]. In the figures we plot 1-BAC because SMAC minimizes a loss function. We will add a paragraph detailing the metrics used in Section 5, but we note that the reported results are qualitatively the same with standard accuracy.

[How do you compute the average ranks?]
Following the evaluation protocols in [16,13], for every dataset we rank the methods at each timestep; we then compute average ranks across datasets.

[Re: intermediary regime in Fig. 3 where the ranks seem more "extreme", and then go back to something more balanced]
Ranks are a relative measurement of performance: the rank of methods 1-4 have to add up to 10 (=1+2+3+4). Thus a method's rank can decrease over time if another method improves. If one method consistently ranks better than another method it outperforms that method in most cases. Meta-learning initially performs much better than no meta-learning, explaining the 'extreme' regime in the middle of the plot. Over time, the Bayesian optimizer SMAC also finds good solutions without meta-learning, letting it catch up on some datasets; this explains that the ranks are less extreme at the end of the optimization. We will explain this in the paper.

Thanks for spotting the 2 typos, we will fix them.

Assigned_Reviewer_2:

[Line 92-93: Could this cross validation still lead to overfitting?]
Yes, using cross-validation error is no silver bullet; there is still a risk of overfitting when evaluating many configurations, but it is smaller than for a single train/test split.

[Any insights into how best to choose the number and type of algorithms to include in the system?]
Yes, we have gathered some informal insights. First, Table 5 suggests that several algorithms are dominated across many datasets and are thus candidates for removal. For the AutoML competition we followed this approach. We removed largely dominated algorithms to speed up optimization and cope with the competition's rigorous constraints on computation time. However, in some cases we found this heuristic to be misleading; finding the 'right' subset of algorithms for a given dataset thus remains an interesting problem for future work.

[Choice of meta-learning]
We chose the presented strategy since it was promising in previous works (e.g. in [13,15]), but we agree that there are several possibilities for performing meta-learning in our system; this is in fact an aspect of the framework we are currently investigating further.

[Line 150: Minor point - how is this Bayesian?]
Sorry, this sentence was misleading; we meant that taking weighted predictions of models with different hyperparameters is closer to the spirit of Bayesian reasoning than relying on a single point estimate. We will clarify this.

Assigned_Reviewer_3:

[Robustness of the method for learning on a novel but very different dataset?]
We agree that assessing the impact of the distribution of datasets on our system, especially on the meta-learning component, is a very interesting question, and we are indeed in the process of evaluating this.

Assigned_Reviewer_6:

[Presentation]
We are sorry that we did not highlight our contributions more clearly upfront; we will do so at the end of the introduction.
We will also describe the Bayesian optimization method we use in more detail. (We note that our methods are not restricted to a specific Bayesian optimizer; we choose SMAC since it yields state-of-the-art performance for CASH problems (see [2,11])).

[Two table numbers are question marks, making the paper seem hurriedly written]
We are sorry for having left that impression. We have indeed made changes to the paper on the day before the deadline, which broke the two references to Table 1. We hope that the remainder of the paper, the thorough experiments and the comprehensive appendix, may partially make up for this oversight and we will of course fix it.

Thanks again to all reviewers!